# SerpinB3/4 Expression Is Associated with Poor Prognosis in Patients with Cholangiocarcinoma

**DOI:** 10.3390/cancers16010225

**Published:** 2024-01-03

**Authors:** Andrea Martini, Kritika Prasai, Tyler J. Zemla, Fowsiyo Y. Ahmed, Mamoun B. Elnagar, Nasra H. Giama, Vincenza Guzzardo, Alessandra Biasiolo, Matteo Fassan, Jun Yin, Patrizia Pontisso, Lewis R. Roberts

**Affiliations:** 1Department of Medicine, University of Padua, via Giustiniani 2, 35128 Padua, Italy; andrea.martini@aopd.veneto.it (A.M.); vincenza.guzzardo@unipd.it (V.G.); alessandra.biasiolo@unipd.it (A.B.); matteo.fassan@unipd.it (M.F.); 2Division of Gastroenterology and Hepatology, Mayo Clinic College of Medicine and Science, 200 First Street SW, Rochester, MN 55905, USA; prasai.kritika@mayo.edu (K.P.); ahmed.fowsiyo@mayo.edu (F.Y.A.); elnagar.mamoun@mayo.edu (M.B.E.); giama.nasra@mayo.edu (N.H.G.); roberts.lewis@mayo.edu (L.R.R.); 3Division of Clinical Trials and Biostatistics, Department of Quantitative Health Sciences, Mayo Clinic College of Medicine and Science, 200 First Street SW, Rochester, MN 55905, USA; zemla.tyler@mayo.edu (T.J.Z.); yin.jun@mayo.edu (J.Y.); 4Veneto Institute of Oncology, (IOV-IRCCS), via Gattamelata 64, 35128 Padua, Italy; 5European Reference Network—ERN RARE-LIVER, 72076 Tübingen, Germany

**Keywords:** SerpinB3/4, cholangiocarcinoma, SerpinB3/4–IgM, prognostic marker

## Abstract

**Simple Summary:**

Cholangiocarcinoma is characterized by a very poor outcome and SerpinB3, a serine protease inhibitor, has recently been found to play a relevant role in malignant transformation in different cancers. The aim of this study was to analyze the expression of this biomarker in the serum and surgical specimens of cholangiocarcinoma in relation to clinical outcome. High levels of SerpinB3/4 were detected in tumoral tissue in 12.2% of CCA, which were characterized by a more advanced TNM stage and lower overall patient survival, independently of CCA subclass. In addition, patients who had detectable free or IgM-linked SerpinB3/4 in serum showed poorer survival. In conclusion, the present study provides evidence that SerpinB3/4, both in the serum and in tumoral tissue, could be considered a useful biomarker to identify the small subgroup of CCA patients with more aggressive tumor biology and dismal prognosis.

**Abstract:**

Cholangiocarcinoma (CCA), the second most common primary liver tumor, is associated with a dismal outcome, and useful prognostic markers are not currently available in clinical practice. SerpinB3, a serine protease inhibitor, was recently found to play a relevant role in malignant transformation in different cancers. The aim of the present study was to determine the expression of SerpinB3/4 in tissue and serum samples of patients with CCA in relation to clinical outcomes. SerpinB3/4 was assessed in the tissue microarrays (TMAs) of 123 surgically resected CCAs. ELISA assays were carried out in 188 patients with CCA to detect the free and IgM-linked forms of SerpinB3/4. Overall survival was analyzed in relation to SerpinB3/4 expression, and Cox models were used to identify the variables associated with survival. High levels of SerpinB3/4 (TMA score 2+/3+) were detected in 15 tumors (12.2%), characterized by a more advanced TNM stage (III/IV: 64.3% vs. 31.3%; *p* = 0.031) and lower overall patient survival, independently of CCA subclass (intrahepatic CCA: median 1.1 (0.8—Not Estimable, NE) vs. 2.4 (1.8–3.4) years; *p* = 0.0007; extrahepatic CCA: median 0.8 (0.2—NE) vs. 2.2 (1.5–5.4) years; *p* = 0.011). Vascular invasion (*p* = 0.027) and SerpinB3/4 scores (*p* = 0.0016) were independently associated with mortality in multivariate analysis. Patients who had detectable free or IgM-linked SerpinB3/4 in their serum showed poorer survival (1 vs. 2.4 years, *p* = 0.015, for free SerpinB3/4, and 1 vs. 2.6 years, *p* = 0.0026, for SerpinB3/4–IgM). In conclusion, high levels of SerpinB3/4 in tissue and serum in CCA are associated with poor outcomes after surgery, regardless of tumor subclass.

## 1. Introduction

Cholangiocarcinoma (CCA) represents the second most common primary liver tumor after hepatocellular carcinoma (HCC), and the overall incidence of CCA, particularly intrahepatic CCA, has increased over recent decades [1,2,3,4,5].

Due to its silent presentation in the early stages, the majority of patients with CCA are diagnosed when the disease is already advanced and unresectable [2,5,6], with a minimal benefit from systemic chemotherapy and a 5-year overall survival (OS) of less than 10% [3,4,5,6,7].

Surgical resection still remains the only potentially curative option for CCA; however, it is feasible for approximately 20% of patients [8,9,10]. Due to the tumor’s aggressiveness and the propensity of CCA to metastasize, even after surgery, the tumor recurrence rate is very high, ranging from 50 to 70%, resulting in unsatisfactory 5-year survival rates (7–20%) [11].

Useful prognostic markers in patients with CCA are still an unmet need, and in the clinical setting, factors such as tumor extent, the presence of metastasis, surgical resection margin and tumor differentiation are currently being used, although they are not ideal for defining patient prognosis [12,13,14]. In the last few years, several molecular markers of prognosis, primarily obtained from serum and tumor tissue, have been proposed as diagnostic and prognostic indicators in CCA. In particular, molecular tissue biomarkers have potential prognostic value, enabling the prediction of patient survival after tumor resection, therapeutic responses and the risk of tumor recurrence, thus allowing patient-tailored therapeutic decisions [15,16,17,18,19,20,21,22]. Different markers of stemness, a very well-known feature of poor outcomes in CCA, have been studied [23,24]; among these, positivity for CD44 and YAP have been documented as tissue markers of worse prognosis [25,26]. Moreover, in different cohorts of resected patients, the presence of genetic alterations of TP53 (tumor protein P53) and KRAS (Kirsten ras oncogene homolog) have been associated with a worse prognosis, characterized by a shorter OS and higher tumor recurrence, compared with in patients carrying isocitrate dehydrogenase (IDH)-1 and -2 mutations [14,21].

SerpinB3 and SerpinB4, also known as squamous cell carcinoma antigen (SCCA)-1 and SCCA-2, respectively, are two highly homologous isoforms belonging to the clade B subset of serine protease inhibitors [27,28]. Recently, the protease inhibitor SerpinB3 has been identified as a critical modulator of the stem-like subset in CCA [29]. In vitro and in vivo experiments have indeed shown that CCA cells that are positive for SerpinB3 have significant features of stemness, pluripotency and Epithelial to Mesenchymal Transition (EMT), which is associated with increased invasiveness and high tumorigenic potential. In addition, patients with iCCA who have had higher SerpinB3 expressions have had significantly lower survival and shorter times to recurrence. Consistently with these results, SerpinB3 has been found to be highly expressed in hepatoblastoma, the embryonal tumor of the liver, particularly in the most aggressive forms, in which a direct correlation has been observed between SerpinB3 gene expression, the upregulation of the Myc oncogene and tumor progression [30]. Further studies have shown that SerpinB3 can increase Myc expression through direct and indirect mechanisms that include the decreased expression of the non-oncogenic Myc-nick cytoplasmic form and the activation of the Yap pathway [31]. Liver tumors with stemness signatures are highly aggressive, and in hepatocellular carcinoma (HCC), the subset of more aggressive tumors, characterized by early recurrence after surgical resection, have shown the highest levels of SerpinB3, associated with high β-catenin and TGF-β1 expression [32]. Additionally, in vitro data have shown that SerpinB3 promotes the EMT and increases cell proliferation and invasiveness [33], while oncogenic Ras upregulates SerpinB3/4 expression, leading to NF-kB activation, IL-6 production and tumor growth [34,35].

SerpinB3/4 isoforms are identifiable in serum through binding to IgM complexes and as free-circulating proteins [36]. SerpinB3/4 linked to IgM is not detectable in healthy subjects, while high or increasing levels of this complex have been described in patients with advanced or worsening liver disease and with increased risk of HCC [37]. In addition, in patients with HCC, high levels of SerpinB3/4–IgM have been associated with shorter survival [38,39].

Although different serum and tissue prognostic molecular biomarkers for CCA have been proposed in recent years, such as miRNAs, SNPs and several signaling molecules, CA19-9 and CEA remain the most clinically used prognostic biomarkers [7], and there is no international consensus on the use of specific molecular tissue biomarkers for patients’ stratification. Thus, the aim of the present study was to analyze the expression of the tissue, free and IgM-complexed blood levels of SerpinB3/4 in human CCA in relation to clinical outcomes.

## 2. Materials and Methods

### 2.1. Patient Analysis

Tissue microarrays (TMAs) created from surgically resected CCAs (*n* = 123) were utilized for this study. For the TMAs, patients who had provided written informed consent and deceased patients who did not opt out of this research, as required by state law, had previously been approved by the Institutional Review Board for inclusion in the Mayo Hepatobiliary Neoplasia Biorepository.

The TMAs were constructed from surplus resected tissue in embedded formalin-fixed paraffin tissue blocks stored in the Institutional Pathology Tissue Archives. Two tumor tissue cores were included per patient. The reading pathologist was blind to the SerpinB3/4 expression.

Clinical data including age at surgery, sex, tumor size, tumor grade, tumor stage, cholangiocarcinoma subtype, vascular invasion, CA19-9, carcinoembryonic antigen (CEA) and alpha-fetoprotein (AFP) were extracted from the electronic medical records (EMRs) for all CCA subjects. CCA was further dichotomized into two groups: extrahepatic CCA (eCCA, which includes perihilar and distal cholangiocarcinoma) and intrahepatic CCA (iCCA).

The serum samples (200 µL each) of an additional 187 patients with CCA (eCCA, 64%; iCCA, 34%) with available clinical outcomes were used for the SerpinB3/4 serum assays. The clinical and histopathologic characteristics of these patients are depicted in Table 1.

### 2.2. SerpinB3/4 Quantification with Immunohistochemistry

Immunohistochemical staining for SerpinB3/4 in the TMAs was performed using the rabbit polyclonal antibody Hepa-Ab (Xeptagen, Venice, Italy). This antibody recognizes both the SerpinB3 and SerpinB4 isoforms, and its specificity has already been confirmed in previous studies and in different oncological settings [30,40]. In brief, 4 μm histological sections, obtained from TMA FFPE blocks, were automatically stained with 5 μg/mL of the Hepa-Ab antibody using a standardized protocol implemented on the automated Leica Microsystems Bondmax^®^ (Leica, Wetzlar, Germany) under the supervision of specialists. Negative and positive controls were included during the staining procedures. An expert gastrointestinal pathologist (MF) scored the cytoplasmic and nuclear staining of the CCA cells for SerpinB3/4 expression while blinded to clinico-pathological information. Immunostaining was semi-quantitated using a four-tier scoring scale based on the intensity of the staining (0 = negative; 1+ = weak; 2+ = moderate; 3+ = strong). Cases were further dichotomized into high-SerpinB3/4 (2+/3+) and low-SerpinB3/4 (0/1+) groups.

### 2.3. Free SerpinB3/4 Determination

Serum SerpinB3/4 was measured using a sandwich ELISA Hepa LISA kit (Xeptagen, Venice, Italy) following the manufacturer’s instructions, as has been previously reported [36]. Briefly, duplicate serum samples were incubated for an hour at room temperature on plates coated with rabbit anti-human-SerpinB3/4 capture antibodies. After washing, the SerpinB3/4 was detected using incubation with HRP-conjugated streptavidin secondary anti-SerpinB3/4 Ab, and the plate was developed with a ready-to-use 3,3′,5,5′-tetramethylbenzidine (TMB) substrate solution. The absorbance at 450 nm was measured on a microplate reader (Victor × 3; Perkin Elmer, Waltham, MA, USA). The cut-off of 3.8 ng/mL was calculated as the value below the 95th percentile on the distribution curve of the assay obtained from the sera of 44 healthy blood donors (male/female ratio = 28/16; age: mean ± SD = 36 ± 9 years).

### 2.4. Determination of SerpinB3/4–IgM

The serum SerpinB3/4 linked to the immunoglobulin M (SerpinB3/4–IgM) immune complexes was measured using an HEPA-IC kit (Xeptagen, Venice, Italy), according to the manufacturer’s instructions, as has been previously described [36]. Briefly, plates precoated with anti-human SerpinB3/4 Ab were incubated with serially diluted standards or serum samples in duplicate. The presence of the SerpinB3/4–IgM complex was revealed with the addition of enzyme-conjugated anti-human IgM. The amount of the immune complex was expressed in Arbitrary Units/mL (AU/mL), and the cut-off value of 200 AU/mL was calculated as the 95th percentile on the distribution curve of the assay for healthy subjects.

### 2.5. Statistical Analysis

We categorized the SerpinB3/4 expression in the tissue as low (score of 0/1+) or high (score of 2+/3+) and compared the clinical characteristics and overall survival (OS) between patients with low and high SerpinB3/4 expression. We summarized the categorical data as frequency counts and percentages and the continuous measures as means, standard deviations, medians and ranges. Categorical variables were compared using the chi-square test or Fisher’s exact test. Continuous variables were compared using the one-way ANOVA test or the Kruskal–Wallis test. For the CCA TMA cohort, OS was defined as the time from the date of surgery to death by any cause; for the serum analysis cohort, OS was defined as the time from diagnosis to death by any cause. The distribution of OS was estimated using the Kaplan–Meier method and compared between patients with high and low SerpinB3/4 expressions using the log-rank test. Hazard ratios (HRs) and 95% confidence intervals (CIs) were estimated using both univariate Cox proportional hazard models and a multivariate Cox model adjusted for confounding factors. Clinical variables with *p*-values of less than 0.2 from the univariate model, including age, sex, albumin, tumor grade, vascular invasion and tumor stage, were included in the multivariate Cox model as confounding factors. All analyses were conducted using two-sided tests with a significance level of 0.05.

## 3. Results

### 3.1. Frequency of SerpinB3/4 Positivity in Human Samples of CCA

We analyzed 123 resected human CCA tumors to evaluate the expressions of SerpinB3/4 in the cancer cell compartments. Fifteen tumors (12.2%) were positive for high levels of SerpinB3/4 (score 2+/3+), and 108 patients (87.8%) had low levels (score of 0/1+) of SerpinB3/4 within the CCA cells. When present in the TMA cores, the normal bile ducts were SerpinB3/4-negative or faintly positive regardless of the background liver condition (Figure 1A). It should be noted that some stromal cells were also positive, particularly plasma cells and rare macrophages, with low/moderate SerpinB3/4 expression. As shown in Figure 1A, in normal livers, no positive stromal cells were detectable, while in the CCA samples, some positive plasma cells were visible in the specimen with a SerpinB3/4 score of 0 (Figure 1B). Examples of the variation in SerpinB3/4 expression in the CCA cells are shown in Figure 1B.

### 3.2. Clinical and Histological Features at Presentation in Relation to the SerpinB3/4 TMA Score

The baseline characteristics of the study population at the time of surgery, including a comparison between patients with high tumor expression of SerpinB3/4 (2+/3+) and patients with low SerpinB3/4 (0/1+), detected with the immunohistochemistry of the TMA scores, are reported in Table 2.

At the time of the surgery, the two groups of patients were similar in age, gender, tumor grade and liver function. Despite the similarity in CCA classification and tumor size between the two groups, patients with high SerpinB3/4 scores (2+/3+) were more frequently classified as TNM stage III/IV (64.3% vs. 31.3%, *p* = 0.031). Moreover, patients with high SerpinB3/4 scores had significantly higher serum CA19-9 levels (328 vs. 53 kU/L, *p* = 0.001). It is worth noting that the patients with high SerpinB3/4 scores (2+/3+) were more frequently found to have micro- or macrovascular invasion, although the differences did not reach statistical significance (13.3% vs. 7.7% and 13.3% vs. 3.8%, respectively).

### 3.3. SerpinB3/4 Score and Prognosis of Cholangiocarcinoma

The CCA patient cohort represented on the TMA was followed for a median of 11.4 (95% CI: 8.7–13.9) years after surgery. Patients with high SerpinB3/4 scores (2+/3+) had significantly lower OS compared to patients with low SerpinB3/4 scores (0/1+) (1.1 years vs. 2.7 years, respectively; *p* < 0.0001).

This probability of survival was significantly lower in the patients with the high scores vs. the low scores even when the patients were categorized by CCA subclass, as shown in Figure 2 (eCCA: median of 0.8 vs. 2.2 years, respectively; *p* = 0.011; iCCA: median of 1.1 vs. 2.4 years, respectively; *p* = 0.0007), or by serum CA19-9 tumor marker level (Appendix A).

### 3.4. Predictors of Overall Survival

The clinical and histological variables that predicted overall survival were analyzed (Table 3).

In a univariate model of OS, sex (*p* = 0.045), age (*p* = 0.17), vascular invasion (*p* = 0.19), albumin (*p* = 0.14) and tumor stage (*p* = 0.11) were significantly prognostic, at a 0.2 significance level (Table 3). When these variables were included in a multivariate model along with SerpinB3/4 scores, only vascular invasion (*p* = 0.027) and the SerpinB3/4 scores (*p* = 0.0016) remained significant (Table 4).

### 3.5. SerpinB3/4 in Serum and Patient Survival

Patients who were positive either for free SerpinB3/4 or for IgM-linked SerpinB3/4 had significantly poorer survival (1 vs. 2.4 years, *p =* 0.015, for free SerpinB3/4, and 1 vs. 2.6 years, *p =* 0.0026, for SerpinB3/4–IgM) (Figure 3).

It should be noted that no correlation was observed between both the free and IgM-linked SerpinB3/4 expression in the serum and tumor grade (*p* = 0.6322 and *p* = 0.2071, respectively, Kruskal–Wallis), as already described in Table 2 of the TMA results (Figure 4).

In addition, the SerpinB3/4 (free or IgM-bound) levels were not different in the groups of patients with the elevated vs. non-elevated CA19-9 levels (Table 5).

## 4. Discussion

Despite recent improvements in understanding the biology of CCA, the clinical outcomes for these patients remains very poor, with rapid progression of the disease after diagnosis and a high frequency of recurrence after surgery [7]. In this context, the identification of reliable prognostic biomarkers, both in tissue and in serum, is an important clinical challenge with this tumor in order not only to personalize therapy but also to define optimal management and to provide the best quality of life and clinical outcome. The protease inhibitor SerpinB3 has been recently identified as a critical modulator of the stem-like subset in CCA [29], and in the present study, while confirming the prognostic role of this biomarker at the tumoral tissue level in a larger series of patients with iCCA and eCCA, we uncovered, for the first time, the prognostic roles of free SerpinB3/4 and SerpinB3/4–IgM in the sera of patients with cholangiocarcinoma after diagnosis. Overall survival was indeed about half as long in patients with detectable levels of this biomarker in the serum or with a high score of expression in tumoral tissue, independently of the CCA subclass. The potential reasons for the detection of SerpinB3/B4 in the subgroup of patients with CCA with a worse prognosis are still unknown and may reside in specific tumor microenvironments, although the precise mechanism thereof deserves further investigation.

Concerning CCA, a great effort has been made in the last few years to identify the different molecular subclasses of the tumor, in particular, with integrative genomic analysis. Sia et al., analyzing 149 iCCAs, identified two molecular subclasses: the inflammation class, characterized by the activation of inflammation related pathways—mainly IL-10, IL-6 and STAT3—and with a more favorable prognosis, and the proliferation class, characterized by the signature of poor prognosis: the activation of the RAS oncogenic pathway and the presence of stem-cell-like iCCA [41]. More recently, an integrative molecular analysis of 189 eCCA tumors has allowed the classification of four molecular subclasses (metabolic, proliferation, mesenchymal and immune) with specific oncogenic profiles [42]. The mesenchymal subclass was dominated by the genomic signals of the EMT, with very intense activation of TGF-β signaling, associated with poor clinical outcomes. It is worth noting that this subclass significantly overlapped with the S1 HCC subclass identified by Hoshida et al. [43], which is characterized by the activation of the TGF-β and WNT pathways, an EMT-like phenotype and a more invasive and metastatic clinical phenotype. Interestingly, the S1 HCC subclass has also been found to express high levels of SerpinB3 [32].

To date, a role for SerpinB3/4 has been documented in many different types of human cancer. In colorectal cancer, SerpinB3, COX-2 and β-catenin have been found to be associated with more advanced tumor stages, and in vitro experiments have also shown that SerpinB3 determines the upregulation of COX-2/ β-catenin positive loops, which is associated with invasive histologic features [44]. In esophageal adenocarcinoma, SerpinB3/4 expression has been associated with reduced tumor chemosensitivity as well as the impairment of immune surveillance, leading to a poor prognosis [45]. Along these lines, SerpinB3 was found to be the most significant response-related gene in melanoma treatment with immune checkpoint blockade therapies and has therefore been proposed to be an independent risk factor in melanoma [46]. In addition, this serpin has been described to drive cancer stem cell survival in glioblastoma [47] and to promote oncogenesis and chemotherapy resistance in breast and ovarian cancer [48,49]. A recent study has also shown that the SerpinB3–Myc axis is upregulated in the basal-like/squamous subtype of pancreatic cancer, known for its aggressiveness [50].

Regarding SerpinB3/4 as a serological biomarker, circulating levels of the free and IgM-linked forms have also been recently investigated in the prediction of the clinical outcomes of esophageal adenocarcinoma and associated with worse overall survival [45]. Those results were in line with those of previous studies, with reports that increased SerpinB3/4–IgM levels were associated with shorter survival not only in patients with HCC [38,39] but also in patients with cirrhosis [37], a known subgroup of patients at higher risk of liver tumor development.

Although CA19-9 levels are also well-described with worsening diagnosis, one of the issues with this biomarker is that about 10–15% of the relevant patients are Lewis-antigen-negative non-producers [51,52]. Since the serum levels of SerpinB3/4 identify patients with worse prognosis, independently of the CA19-9 values, this biomarker could overcome the problem and improve the management of these patients.

This study had some limitations, such as the absence of a validation group and the limited number of patients found positive for SerpinB3/4 in tumoral tissue and in the serum. It is worth noting that the external validation groups from the different international cohorts that confirmed our data had been previously evaluated in the recent study of Correnti et al. [29]. Regarding the low percentage of patients positive for SerpinB3/4 in tissue and in the serum, these results are in line with the percentage of well-known prognostic molecular alterations recently studied in a real-world cohort of patients with biliary tract cancer [53], although the recently described analysis of a proteomic profile of circulating extracellular vesicles could open new scenarios and improve the identification of patients at risk of CCA development [54].

## 5. Conclusions

In conclusion, the present study provides evidence that SerpinB3/4, both in the serum and in tumoral tissue, could be considered a useful biomarker to identify the small subgroup of CCA patients with more aggressive tumor biology and dismal prognosis. The usefulness and the significance of these biomarkers need further evaluation in future studies, possibly in combination with other markers, to better tailor the clinical management of these patients.

## Figures and Tables

**Figure 1 cancers-16-00225-f001:**
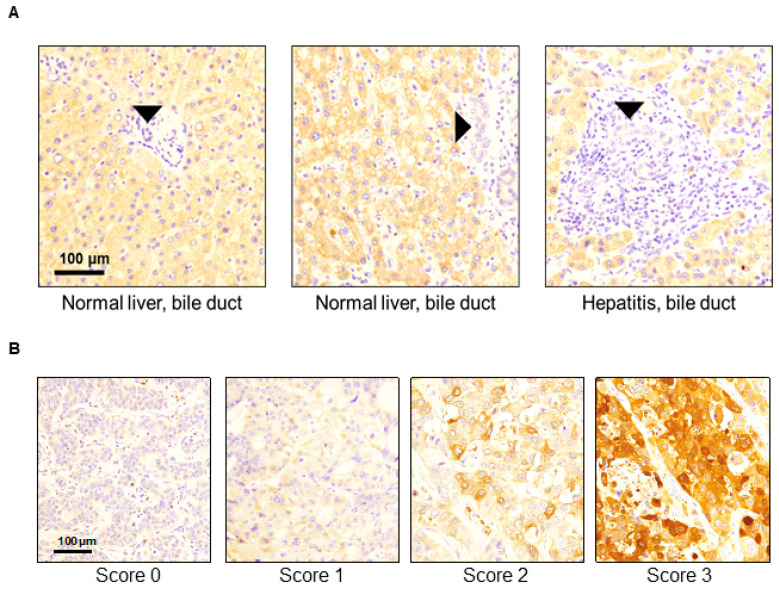
Representative examples of SerpinB3/4 immunohistochemical expression in TMA samples obtained from surgically resected cholangiocarcinomas. (**A**) Normal bile ducts (black arrows’ heads) in non-neoplastic liver parenchyma showed no/faint SerpinB3/4 expression. (**B**) Representative SerpinB3/4 expression (brown color) in CCA samples according to the IHC expression scores.

**Figure 2 cancers-16-00225-f002:**
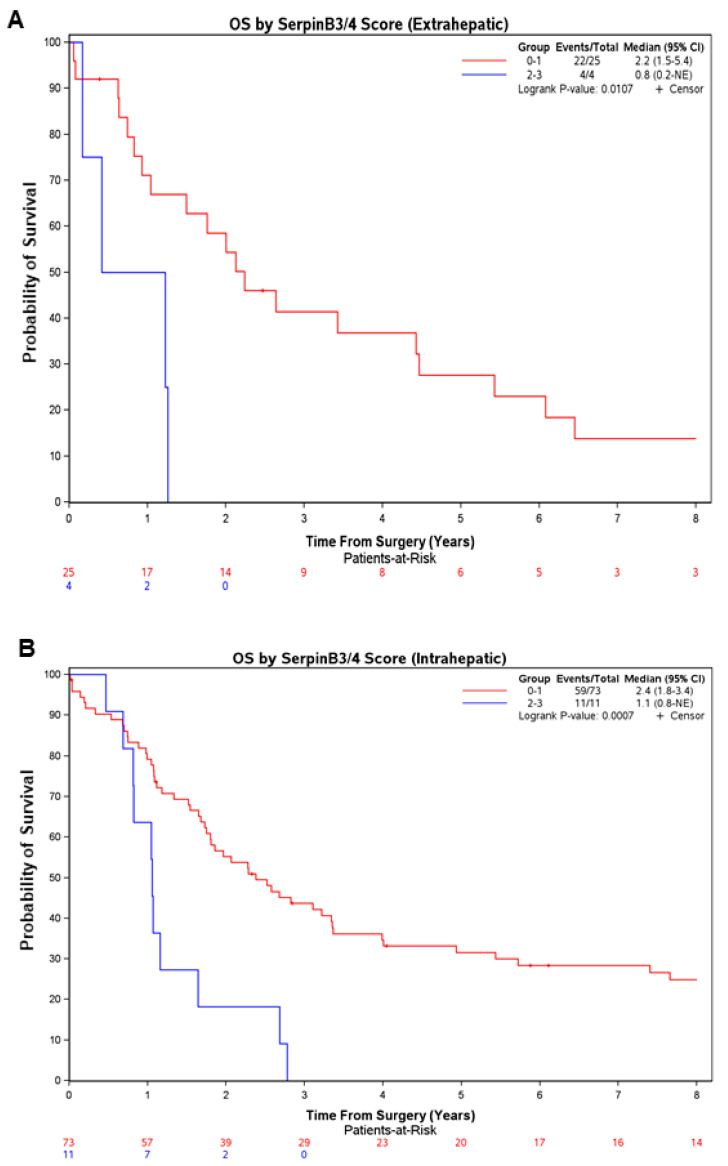
Patients with higher immunohistochemistry scores for SerpinB3/4 in resected CCA had lower overall survival regardless of the CCA subtype. Overall survival after surgical resection in the CCA cases (eCCA in (**A**); iCCA in (**B**)) subclassified by the SerpinB3/4 score determined with immunohistochemistry (*n* = 123) (high SB3 = score of 2–3, low SB3 = score of 0–1). Survival curves were estimated using the Kaplan–Meier method, and differences between curves were assessed with a long-rank test. CI, confidence interval; NE, Not Estimable.

**Figure 3 cancers-16-00225-f003:**
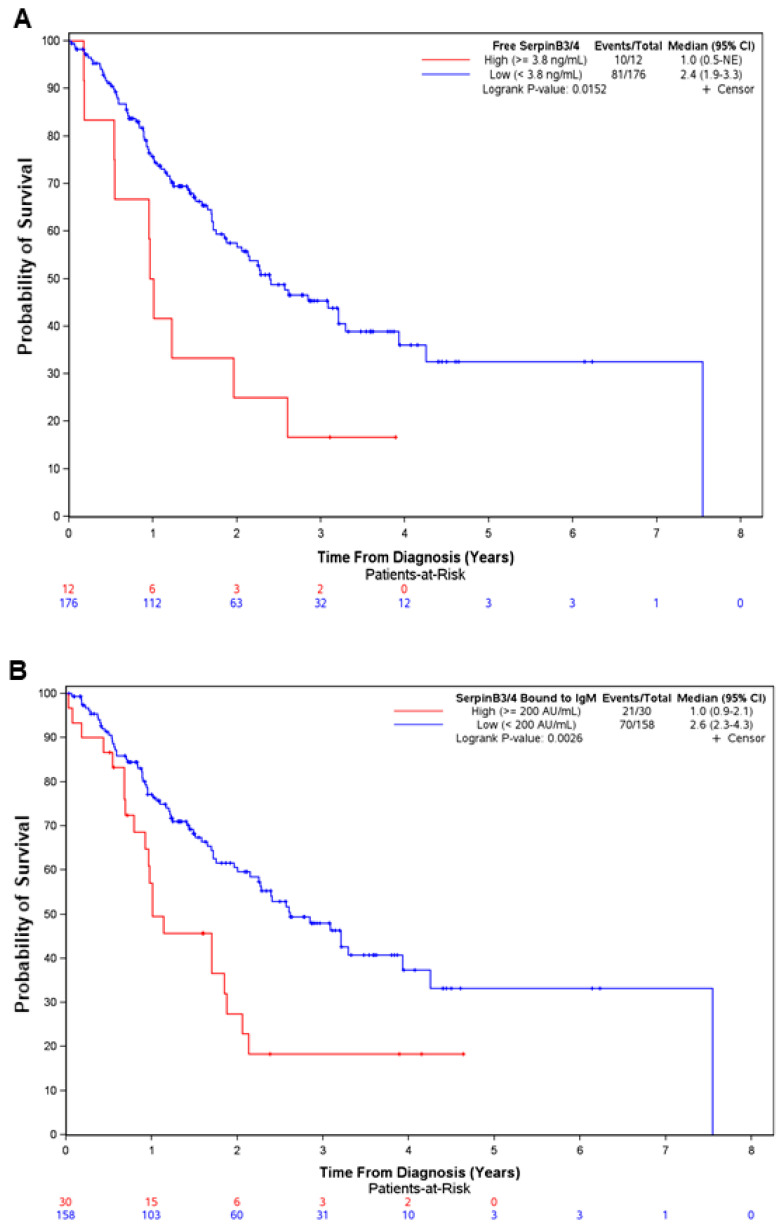
Positivity for free SerpinB3/4 and SerpinB3/4–IgM in serum predicts poor survival in patients with CCA. Overall survival from diagnosis in 188 patients with CCA, divided according to the positivity from two different SerpinB3/4 serum assays (free SerpinB3/4, (**A**); SerpinB3/4–IgM, (**B**). Survival curves were estimated using the Kaplan–Meier method, and the differences between curves were assessed with a long-rank test. CI, confidence interval; NE, Not Estimable.

**Figure 4 cancers-16-00225-f004:**
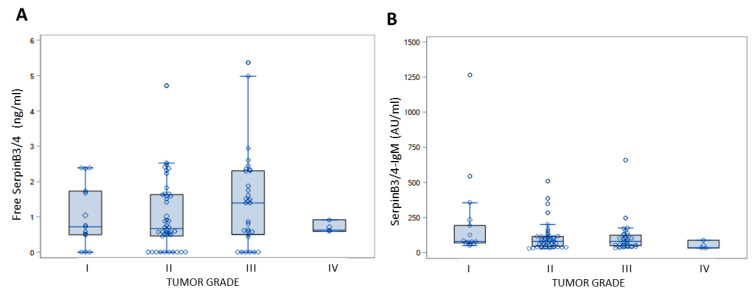
Serum levels of free SerpinB3/4 and SerpinB3/4–IgM in in relation to tumor grade in patients with CCA. Box plot results of serum levels of free SerpinB3/4 (**A**) and of IgM-linked SerpinB3/4 (**B**) in patients with CCA, grouped by tumor grade. The box indicates the lower and upper quartiles, the middle line indicates the median and the diamond indicates the mean. I to IV refer to tumor grade.

**Table 1 cancers-16-00225-t001:** Clinical and histopathologic characteristics of the patients with cholangiocarcinoma used for testing SerpinB3/4 in serum.

Total Number of Patients	187
**SEX**	
Female	29.9%
Male	70.1%
**AGE**	
Mean (SD)	58.8 (13.2)
Median	61.0
Q1, Q3	49.0, 69.0
Range	(22.0–87.0)
**CCA CLASSIFICATION**	
Extrahepatic	64.0%
Intrahepatic	36.0%
**ALP**	
Mean (SD)	397.7 (376.6)
Median	302.0
Q1, Q3	130.0, 524.0
Range	(62.0–2154.0)
**BILIRUBIN**	
Mean (SD)	4.5 (6.1)
Median	1.8
Q1, Q3	0.7, 5.8
Range	(0.2–32.3)
**CA19-9**	
Mean (SD)	4332.5 (22,423.5)
Median	125.0
Q1, Q3	30.0, 441.0
Range	(1.0–21,1100.0)
**CEA**	
Mean (SD)	523.0 (3212.7)
Median	2.5
Q1, Q3	1.3, 6.6
Range	(0.3–26,630.0)
**TNM STAGE**	
I	22.8%
II	16.5%
III	27.8%
IV	32.9%
**TUMOR GRADE**	
I	9.6%
II	55.7%
III	31.1%
IV	3.6%

ALP: alkaline phosphatase; CCA: cholangiocarcinoma, CEA: carcinoembryonic antigen; TNM: tumor, node, metastasis.

**Table 2 cancers-16-00225-t002:** Baseline characteristics of the patients at the time of surgery in relation to SerpinB3/4 TMA scores.

	Total (N = 123)	Score 0/1+ (N = 108)	Score 2+/3+ (N = 15)	*p*-Value
**Sex**				0.7843 ^1^
Female	56 (45.6%)	50 (46.3%)	6 (40.0%)	
Male	67 (54.4%)	58 (53.7%)	9 (60.0%)	
**Age**				0.0897 ^2^
Median (IQR)	64.0 (55, 71)	63.0 (54, 69)	67.0 (60, 75)	
**TB (mg/dL)**				0.8836 ^2^
Median (IQR)	0.6 (0.5, 0.9)	0.7 (0.5, 0.9)	0.6 (0.5, 1.6)	
**ALB (gr/dL)**				0.7523 ^2^
Median (IQR)	4.2 (3.7, 4.4)	4.2 (3.7, 4.4)	4.2 (3.6, 4.4)	
**ALT (U/L)**				0.4651 ^2^
Median (IQR)	4.4 (2.5, 8.1)	4.5 (2.5, 8.0)	3.3 (2.0, 9.2)	
**PT (s)**				0.0442 ^2^
Median (IQR)	9.6 (9.1, 10.5)	9.6 (9.2, 10.7)	9.2 (7.9, 9.8)	
**CEA (** **μ** **g/L)**				0.8958 ^2^
Median (IQR)	1.6 (0.9, 2.6)	1.6 (0.9, 2.6)	1.4 (1.4, 2.9)	
**CA19-9 (kU/L)**				0.0064 ^2^
Median (IQR)	68 (18, 289)	50 (17, 239)	311 (159, 1112)	
**CCA Classification**				0.8736 ^1^
Intrahepatic	74.3%	74.5%	73.3%	
Perihilar	21.2%	20.4%	26.7%	
Distal	4.4%	5.1%	0.0%	
**Tumor Size (cm)**				0.9457 ^2^
Median (IQR)	5.0 (3.3, 8.0)	5.0 (3.2, 8.0)	5.2 (3.5, 8.5)	
**TMN Stage**				0.0318 ^1^
I/II	60 (63.8%)	55 (68.8%)	5 (35.7%)	
III/IV	34 (36.2%)	25 (31.3%)	9 (64.3%)	
**Tumor Grade**				1.0000 ^1^
I/II	21 (17.5%)	19 (18.1%)	2 (13.3%)	
III/IV	99 (82.5%)	86 (81.9%)	13 (86.7%)	
**Vascular Invasion**				0.0909 ^1^
None	86.6%	88.5%	73.3%	
Micro	8.4%	7.7%	13.3%	
Macro	5.0%	3.8%	13.3%	

TB, total bilirubin; ALB, albumin; ALT, alanine aminotransferase; PT, Prothrombin Time; CEA, carcinoembryonic antigen; CA19-9, Gastrointestinal Cancer Antigen; CCA, cholangiocarcinoma; TNM, tumor size, nodes, metastasis. ^1^ Fisher’s exact test; ^2^ Kruskal–Wallis.

**Table 3 cancers-16-00225-t003:** Clinical and histological variables in relation to overall survival, analyzed with a univariate overall survival model.

Variable	Level	HR (95% CI)	*p*-Value (vs. Reference)	*p*-Value
**Sex (Ref = Male)**	Female	0.66 (0.44, 0.99)	0.044	0.044
**Age**		1.01 (0.99, 1.03)	0.170	0.170
**Tumor Grade** (Ref = I/II)	III/IV	1.30 (0.75, 2.279)	0.345	0.345
**Vascular Invasion** (Ref = None)	Micro	1.71 (0.78, 3.75)	0.180	0.187
**Vascular Invasion** (Ref = None)	Macro	1.80 (0.77, 4.22)	0.171	0.187
**AFP**		0.99 (0.94, 1.04)	0.755	0.755
**TB**		1.05 (0.97, 1.13)	0.208	0.208
**ALB**		0.89 (0.76, 1.03)	0.140	0.140
**ALT**		0.98 (0.95, 1.02)	0.497	0.497
**PT**		1.00 (0.93, 1.06)	0.997	0.997
**CCA Classification** (Ref = Intrahepatic)	Perihilar	1.18 (0.72, 1.92)	0.512	0.630
**CCA Classification** (Ref = Intrahepatic)	Distal	1.43 (0.57, 3.61)	0.438	0.630
**Tumor Size** (cm)		1.02 (0.97, 1.08)	0.313	0.313
**Tumor Stage** (Ref = I/II)	III/IV	1.47 (0.91, 2.37)	0.113	0.113

Univariate COX PH Modeling (OS anchored at surgery date). AFP, alpha-fetoprotein; TB, total bilirubin; ALB, albumin; ALT, alanine aminotransferase; PT, Prothrombin Time; CEA, carcinoembryonic antigen; CA19-9, Gastrointestinal Cancer Antigen.

**Table 4 cancers-16-00225-t004:** Clinical and histological variables in relation to overall survival, analyzed with a multivariate overall survival model.

Factor	Level	HR (95% CI)	*p*-Value
**SerpinB3/4 Score**	2+/3+ vs. 0/1+	3.23 (1.59, 6.55)	0.001
**Age**	Unit = 5	1.01 (0.90, 1.13)	0.920
**Sex**	Female vs. Male	0.72 (0.43, 1.21)	0.211
**Albumin**	Unit = 1	0.95 (0.78, 1.16)	0.615
**Vascular Invasion**	Micro vs. None Macro vs. None	1.76 (0.74, 4.20) 4.55 (1.29, 16.04)	0.201 0.018
**Tumor Stage**	Stage III/IV vs. Stage I/II	1.58 (0.94, 2.69)	0.086

Multivariate COX PH modeling (OS anchored at surgery date). OS, overall survival; HR, hazard ratio; CI, confidence interval.

**Table 5 cancers-16-00225-t005:** Serum levels of free and IgM-bound SerpinB3/4 in patients grouped in relation to the serum levels of CA19-9.

	CA19-9 Non-Elevated (N = 30)	CA19-9 Elevated (N = 76)	Total (N = 106)	*p*-Value
**Free SerpinB3/4 (ng/mL)**				0.4384 ^1^
N	30	76	106	
Mean (SD)	1.2 (1.0)	1.1 (1.1)	1.1 (1.1)	
Median	0.9	0.7	0.7	
Q1, Q3	0.6, 1.7	0.5, 1.8	0.5, 1.8	
Range	(0.0–4.7)	(0.0–5.4)	(0.0–5.4)	
**SerpinB3/4–IgM (AU/mL)**				0.2588 ^1^
N	30	76	106	
Mean (SD)	98.1 (108.4)	150.0 (231.5)	135.3 (205.1)	
Median	73.9	76.2	75.9	
Q1, Q3	52.2, 93.6	50.5, 136.4	50.5, 125.2	
Range	(30.1–544.5)	(29.6–1482.2)	(29.6–1482.2)	

CA19-9, Gastrointestinal Cancer Antigen.^1^ Kruskal–Wallis.

## Data Availability

The data presented in this study are available upon request. These data are not publicly available due to privacy reasons.

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
