# Peer review of "SerpinB3/4 Expression Is Associated with Poor Prognosis in Patients with Cholangiocarcinoma"

_cancers, 2024, doi:10.3390/cancers16010225_

Round 1
Reviewer 1 Report (Previous Reviewer 2)
Comments and Suggestions for Authors
The authors have addressed all comments satisfactorily.
Author Response
We are grateful to the Reviewer for his overall positive consideration of the revised manuscript.
Reviewer 2 Report (Previous Reviewer 1)
Comments and Suggestions for Authors
Authors addressed issues.
Comments on the Quality of English LanguageSome sentences and grammar needs to be checked prior publication.
Author Response
We are grateful to the Reviewer for his recommendation and some sentences and grammar have been modified, as suggested.
Reviewer 3 Report (New Reviewer)
Comments and Suggestions for Authors
In the Manuscript “SerpinB3/4 expression is associated with poor prognosis in patients with cholangiocarcinoma“ by Martini et al., the authors investigated if levels of expression of SerpinB3/4 in tissue and serum samples of patients with CCA correlate with clinical outcome. They analyzed Serpin B3/4 expression in tumor tissues (eCCA and iCCA) using tissue microarrays (TMAs) of assembled of 123 surgically resected CCAs and in 188 CCA patient serums using ELISA assay (to detect free and IgM linked forms of SerpinB3/4). Moreover, the authors analyzed Overall Survival relative to SerpinB3/4 expression and used Cox models to identify variables associated with survival. The major limitation of the study is low number of patient samples especially relative to TMA staining and group with high SerpinB3/4 expression (only 15). This limitation is recognized by the authors in the Conclusion section. The study is focused and of significant clinical interest since there is an unmet need for discovery of more useful prognostic markers in patients with CCA.
Specific comments:
1. Line 144: please explain MF.
2. Figure 1: images taken at higher magnification would be useful. It is not clear from the figure which cells are expressing SerpinB3/4. In the tumor samples with high SerpinB3/4 expression, is it possible to address type of the positive cells? Are mainly tumor cells positive or some stromal cells as well? Is SerpinB3/4 intracellular or extracellular? This is an important question and it should be addressed in the manuscript.
3. Table 2: 2 rows are marker with “AGE, Median (IQR). Is this a mistake? Also, the same row, column “Total” in parenthesis is missing comma (,).
Author Response
We are grateful to the Reviewer for his hoverall consideration of the revised manuscript.
The reply to the specific comments is following.
- The acronym MF refers to Matteo Fassan, full Professor of Pathology at the Dept of Medicine of the University of Padua, who is one of the authors of the present manuscript.
- We appreciate the comment of the reviewer for the opportunity provided to specify that we focused on SerpinB3/4 expression at the cholangiocyte/epithelial cancer cells level and this detail has been added to the Methods Section. Some stromal cells were also positive, particularly plasma cells and rare macrophages, with low/moderate SerpinB3/4 expression. As shown in Figure 1A, in normal liver, no positive stromal cells were detectable, while in Figure 1B of cholangiocarcinoma samples, some positive plasma cells were visible in the specimen with score 0. Of note, for the survival analysis, only the score of SerpinB3/4 within the tumor cells was considered, and the reported data confirm that the intracellular expression of this serpin in neoplastic cells provides a negative prognostic value. This information has been added to the Results Section.
- We apologize for this mistake that has been corrected in the revised Table 2.
This manuscript is a resubmission of an earlier submission. The following is a list of the peer review reports and author responses from that submission.
Round 1
Reviewer 1 Report
Comments and Suggestions for Authors
In the paper submitted to MDPI cancer titled: SerpinB3/4 expression is associated with poor prognosis in patients with cholangiocarcinoma by Andrea Martini et al., authors analyze 123 samples from human with Cholangiocarcinoma for presence of Serpin B3/4 using TMA and analyze serum of 188 patients using ELISA for SERPIN B3/4. In general, paper is well organized however is missing data representation that would support and connect two parts of this paper, and future use of serpinB3/4 in a future as a biomarker. Major revisions are recommended prior publication.
Following questions should be answered prior publication.
1. Please provide information on the control group of patients used in ELISA assay. Addition of graph representing the level of detectable SerpinB3/4 immunoglobulin bound or free in serum would illustrate how high levels of serpinB3/4 correlate with disease progression and link serum of 188 patients analyzed via ELISA with cholangiocarcinoma of 123 patients analyzed via TMA. A simple graph of expression level with the grade of the tumor at the time of blood collection would be helpful.
2. In Materials and Methods please explain in detail how TMA were analyzed by a pathologist.
Author Response
We are grateful to the reviewer for his overall consideration of our manuscript. Please find below the reply to the points addressed in his review:
Please provide information on the control group of patients used in ELISA assay.
The demographic characteristics of the control group used in the ELISA assay has been added in the Methods section.
Addition of graph representing the level of detectable SerpinB3/4 immunoglobulin bound or free in serum would illustrate how high levels of serpinB3/4 correlate with disease progression and link serum of 188 patients analyzed via ELISA with cholangiocarcinoma of 123 patients analyzed via TMA. A simple graph of expression level with the grade of the tumor at the time of blood collection would be helpful.
We are grateful to the Reviewer for his important suggestion and a new Figure 4 has included in the revised version. As observed by TMA, also the levels of free and IgM-bound SerpinB3/4 in serum were not related to tumor grade. This information has been added to the Results section.
In Materials and Methods please explain in detail how TMA were analyzed by a pathologist.
More information has been added in the Materials and Methods, as requested.The sentence was modified as follows “An expert gastrointestinal pathologist (MF) scored cytoplasmic and nuclear staining for SerpinB3/4 expression blinded to clinico-pathological information”.

Reviewer 2 Report
Comments and Suggestions for Authors
In this manuscript by Martini, Prasai and colleagues, the authors demonstrate the prognostic implications of SerpinB3/B4 levels in tissue and serum from patients with CCA. This study reports impressively sized cohorts for this rare disease, totaling 123 tissues and 187 serum samples. The experiments have been well conducted; the methods and analyses are well described; overall, the manuscript is well written. Collectively, this study builds a compelling case for the potential clinical utility of measuring SerpinB3/B4 levels.
I only have minor comments:
- The authors note that high IgM-bound SerpinB3/B4 are associated with higher risk of HCC and prognosis (in the introduction). In the CCA cohort, is there any association between etiological status (cirrhosis, hepatitis, primary sclerosing cholangitis) and SerpinB3/B4 levels (staining intensity in tissues, abundance of free or IgM-bound SerpinB3/B4 in circulation)?
- The authors find high SerpinB3/B4 expression in tissues to be associated with higher serum CA19-9 levels, the latter being well described with worsening prognosis in CCA. However, one of the issues with CA19-9 is that ~10-15% are Lewis antigen-negative non-producers. Can the authors demonstrate that combining CA19-9 with SerpinB3/B4 (free or IgM-bound) improves the prognostic performance of CA19-9 in multivariate analysis? Alternatively, can the authors demonstrate that serum SerpinB3/B4 levels are not different between patients with non-elevated CA19-9 (presumed Lewis antigen-negative) versus the remainder of the cohort?
- Space-permitting, the discussion might benefit from mentioning:
· Potential reasons for SerpinB3/B4 detection in a subgroup of CCA (for example, increased apoptotic cell death in specific tumor microenvironments)
· The potential utility of monitoring SerpinB3/B4 during chemotherapy as a non-invasive metric for disease control (particularly in Lewis antigen-negative patients)
- Line 53: “metastasis” should be “metastasise” or “metastasize” (depending on UK versus US English).
Author Response
We appreciate the overall consideration of the Reviewer for our study. The reply to his comments is listed below:
- The authors note that high IgM-bound SerpinB3/B4 are associated with higher risk of HCC and prognosis (in the introduction). In the CCA cohort, is there any association between etiological status (cirrhosis, hepatitis, primary sclerosing cholangitis) and SerpinB3/B4 levels (staining intensity in tissues, abundance of free or IgM-bound SerpinB3/B4 in circulation)?
The point raised by the Reviewer is interesting, but unfortunately, we do not have adequate data collection of the etiological status in this population at this time to determine if there is any association with SerpinB3/4 levels.
- The authors find high SerpinB3/B4 expression in tissues to be associated with higher serum CA19-9 levels, the latter being well described with worsening prognosis in CCA. However, one of the issues with CA19-9 is that ~10-15% are Lewis antigen-negative non-producers. Can the authors demonstrate that combining CA19-9 with SerpinB3/B4 (free or IgM-bound) improves the prognostic performance of CA19-9 in multivariate analysis? Alternatively, can the authors demonstrate that serum SerpinB3/B4 levels are not different between patients with non-elevated CA19-9 (presumed Lewis antigen-negative) versus the remainder of the cohort?
We are grateful to the Reviewer for this important consideration. Although also CA19-9 levels are well described with worsening diagnosis, in the new Table 5 is shown that SerpinB3/4 (free or IgM-bound) levels are not different in the group of patients with elevated vs non-elevated CA19-9 levels.
- Space-permitting, the discussion might benefit from mentioning: Potential reasons for SerpinB3/B4 detection in a subgroup of CCA (for example, increased apoptotic cell death in specific tumor microenvironments); the potential utility of monitoring SerpinB3/B4 during chemotherapy as a non-invasive metric for disease control (particularly in Lewis antigen-negative patients).
We are grateful for providing us the possibility to expand and improve the Discussion section addressing the points raised by the reviewer.
- Line 53: “metastasis” should be “metastasise” or “metastasize” (depending on UK versus US English).
The mistake has been corrected.
